# Sucralose-Enhanced Adipogenesis on Preadipocyte Human Cell Line During Differentiation Process

**DOI:** 10.3390/ijms252413635

**Published:** 2024-12-20

**Authors:** Javier A. Magaña-Gómez, Guadalupe González-Ochoa, Jesus A. Rosas-Rodríguez, N. Aurora Stephens-Camacho, Lilian K. Flores-Mendoza

**Affiliations:** 1Posgrado en Ciencias Biomédicas, Facultad de Ciencias Químico Biológicas, Universidad Autónoma de Sinaloa, Av. de las Américas y Josefa Ortiz, CU 80010 Culiacán, Sinaloa, Mexico; jmagana@uas.edu.mx; 2Laboratorio Universitario de Análisis Clínicos e Investigación, Universidad de Sonora (LUACI) Departamento de Ciencias Químico-Biológicas y Agropecuarias, Campus Navojoa. Lázaro Cárdenas del Río #100, CP 85880 Navojoa, Sonora, Mexico; guadalupe.gonzalezochoa@unison.mx (G.G.-O.); jesus.rosas@unison.mx (J.A.R.-R.); 3Licenciatura en Nutrición Humana, Universidad Estatal de Sonora, Blvd. Manlio Fabio Beltrones 810, Col. Bugambilias, CP 85875 Navojoa, Sonora, Mexico

**Keywords:** white adipocyte, adipogenesis, lipogenesis, inflammation, gene expression, sucralose, nutrition, adipose tissue, inflammation, obesity

## Abstract

Sucralose, a commonly nonnutritive sweetener used in daily products of habitual diet, is related to impairing the gut microbiome by disrupting inflammatory response, promoting weight gain by increasing adipose tissue and promoting chronic inflammatory processes. Considering the impact of sucralose in the development of metabolic diseases, in this work, we focused on the impact of sucralose on the adipocyte differentiation process to determine if sucralose can promote adipogenesis and increase adipose tissue depots in PCS 210 010 human preadipocytes cell line. Sucralose at 25 (S25) and 100 ng/µL (S100) concentrations were tested against control with no edulcorant (NS) during the adipocyte differentiation process at 48 h and 96 h. The genetic expression of adipogenesis markers such as CEBP-α, PPARγ, EBF-2, UCP-1, and lipogenesis regulator ACC was determined by qPCR. A panel of human cytokines related to inflammatory response was measured by a flow cytometer using the kit Legend Plex Human Cytokine panel of BIOLUMINEX. Our results indicate that sucralose increased the expression of white adipocyte differentiation marker CEBP-α and lipogenesis regulator ACC at 96 h before complete differentiation. Also, sucralose triggers an inflammatory response by synthesizing adiponectin, resistin, IL-6, IL-8, and Il-1B. To summarize, sucralose stimulates the expression of genes related to adipogenesis and negatively affects the secretion of inflammatory cytokines and adipokines during preadipocyte differentiation.

## 1. Introduction

The prevalence of obesity worldwide is a concerning issue in population health because comorbidities such as diabetes and metabolic syndrome are derived [1]. These conditions are provoked mainly by a positive energy balance in fatty tissue increases due to excessive calorie intake. Because of this, the food industry is trying to prevent obesity-related diseases and different food alternatives have been developed, such as non-caloric sweeteners (NNS) [2]. Sucralose is the most widely NNS present in sugar-free and low-sugar products due to generally being recognized as safe (GRAS) by the Food and Drug Administration (FDA) [3,4,5,6,7]. However, sucralose consumption is controversial because of its effects on metabolic levels, weight gain, and gut microbiota homeostasis, related to increased adipose tissue depots in rats and another animal model, showing an association between consumption and metabolic disorders [8,9,10,11]. The potential implications of sucralose on obesity and metabolic syndrome are significant, and our research aims to shed light on them. The adipose tissue is conformed mainly by white adipocytes and, in minor proportions, preadipocytes and mesenchymal cells, responsible for generating more adipose tissue in response to external stimuli through the adipocyte differentiation process [12]. Adipocyte differentiation usually occurs during adipocyte life to replenish adipocyte number; it also occurs in response to an environmental stimulus when energy storage in triglycerides is needed, increasing the number and size of adipocytes [13]. This adipocyte differentiation, triggered by the expression of lipogenic genes like CEBP-α and PPARγ, promotes the increase in the depot of adipose tissue and increases the synthesis of inflammatory cytokines related to obesity, such as IL-6, TNF-aα, and IFN-γ, a critical factor in developing degenerative and chronic diseases like obesity and metabolic syndrome. Sucralose in concentrations of 0.1 mM increases the expression of adipogenic genes in mesenchymal cells derived from human adipose tissue [5]. The perception of sucralose by adipocytes is mediated by membrane receptors accoupled to g protein, conformed by TAS1R2 and TAS1R3 subunit. Our work focuses on sucralose’s effect at 48 and 96 h during the adipocyte differentiation process in the human preadipocyte cell line, measuring the genetic expression of adipogenesis and lipogenesis markers and cytokine-adipokine synthesis.

## 2. Results

### 2.1. Effects of Sucralose on Adipocyte Morphology

At the early stage of differentiation (48 h), adipocytes exhibited a fibroblast-like morphology in both sucralose concentrations. However, at complete differentiation (96 h), we observe a higher number of differentiated cells with adipocyte-like morphology with S25 stimuli than adipocytes stimulated with S100 concentration and those without sucralose (NS) (Figure 1). Furthermore, none of the tested concentrations showed cytotoxic effects on differentiated adipocytes.

### 2.2. Expression of White/Brown Adipocyte Gene Markers and Lipogenesis

We found that sucralose did not increase the genetic expression of lipogenesis and differentiation markers at the early stages of differentiation. At the early differentiation process at 48 h, the expression of brown and white genetic markers was low for both sucralose stimuli concerning controls. At the final differentiation process at 96 h on brown adipocyte differentiation, markers showed similar behavior in EBF-2 and UCP-1. However, it is essential to mention that at 96 h, S25 stimuli increases 10X the genetic expression of white adipocyte differentiation markers as CEBP-α and 1.8X ACC synthase related to lipogenesis.

The lowest expression values in CEBP-α and PPARγ were observed in the first 48 h after starting the adipocyte differentiation process. However, the CEBP-α gene was overexpressed 30-fold concerning the control NS at the end of the differentiation (96 h) under S25 stimuli. The concentration S25 is statistically different (*p* < 0.0001) between both 48 h and 96 h differentiation time and the sucralose S100 concentration at the end of the differentiation. The PPARγ also increased its expression at the end of the adipocyte differentiation, showing a statistical difference between both evaluated times (*p* = 0.0065) (Figure 2A,B). However, there was no significance between S25 and S100 sucralose concentrations in the tested two differentiation times 48 h and 96 h.

Additionally, the expression results for the marker genes differentiate brown adipocytes EBF-2 and the uncoupling protein UCP-1, showing that there is an increase in the expression of EBF-2 and UCP-1 at 96 h with significance between both differentiation times (*p* < 0.0001). However, these brown adipocyte markers’ increased expression levels are not similar to our values obtained for CEBP-α. Our results show that the concentration S25 favored the increase in the expression of EBP-2 and UCP-1 at 96 h of the adipocyte differentiation process (Figure 2C,D).

In terms of regulating lipogenesis, it was noticed that both sucralose S25 and S100 caused an increase in the expression of ACC after 96 h of adipocyte differentiation. This effect was more noticeable with the S25 concentration of sucralose, compared to 48 h and each other (*p* < 0.0001) (Figure 2E).

### 2.3. Human Adipokine Response to Sucralose Stimuli on Adipocyte Differentiation by Flow Cytometry

We quantified five adipokines (adiponectin, leptin, adipsin, resistin, and RBP4) and MCP-1 chemokine at 48 h (early differentiation) and 96 h (complete differentiation) under the stimulus of sucralose 25 and 100 ng/µL (S25 and S100) and the control without sucralose (NS).

Adiponectin levels on NS at 48 h were 2558.13 pg/mL; the sucralose stimuli decreased the levels of this adipokine significantly with the stimulation of S25 (1277.18 pg/mL, *p* = 0.002) and not significantly with the S100 stimulus. This decrease in adiponectin levels with sucralose stimulation could indicate a potential adverse effect of sucralose on adipocyte differentiation. At 96 h, with the differentiation process finalizing, the adiponectin levels were 1863.67, 1483.12, and 1790.54 pg/mL for NS, S25, and S100, respectively, with no statistical differences between treatments concerning the early stage of differentiation at 48 h. These results suggest that the effects of sucralose on adiponectin levels may be more pronounced at the early stages of differentiation.

Leptin levels exhibit a similar pattern, with lower values for NS, 633.86 pg/mL, and a decrease to 446.97 pg/mL with S25, a significant difference with a *p*-value of 0.02. No statistical difference was found in leptin levels at 96 h (Figure 3B).

Regarding adipsin, the levels of this adipokine were 6308.3 pg/mL for NS and 48 h with no statistical difference from NS at 96 h; similar results were obtained for S25. However, for S100, the levels of adipsin at 48 h were comparable to the control and did not show a statistical difference; however, at 96 h, S100 had a significative increase with an adipsin level of 18,780.4 pg/mL (*p* = 0.01); this increase is significantly different between times (48 h and 96 h), *p* = 0.005. Furthermore, a significant difference was found between the stimuli with sucralose (S25 and S100) with *p* = 0.001 (Figure 3C).

The effect of sucralose stimulus on resistin concentration was remarkable during the first 48 h of differentiation. This adipokine’s concentration values were 161.4, 99, and 107 pg/mL for NS, S25, and S100 at 48 h of differentiation. At the early differentiation process, we observe a 38% decrease in resistin levels with the S25 stimulation and 33% with the S100 stimulus, statistically different from NS, showing *p* = 0.01 and *p* = 0.03 values, respectively. At 96 h, the resistin values are diminished by 30.5% with a concentration of 112 pg/mL, showing statistical differences between 48 h and 96 h of differentiation (*p* = 0.02). No stimulus showed statistical differences at 96 h of differentiation (Figure 3D).

For RBP4, we observed no significant differences between stimuli in the early differentiation time (48 h); 13,173, 12,472, and 13,960 pg/mL values were obtained for NS, S25, and S100, respectively. However, at 96 h, S25 and S100 increased the production of RBP4 concerning NS control, with a statistical difference for S25 (17,360 pg/mL, *p* = 0.002).

The last compound quantified was the chemokine MCP-1 (monocyte chemoattractant protein). The results show that in NS at 96 h, the concentration increased significantly (*p* = 0.0001), 2.8-fold from 875 pg/mL quantified at 48 h to 2497 pg/mL at 96 h. However, at 96 h, stimulation with S25 and S100 decreased the concentration of MCP-1 by approximately 80%, from 595.3 and 691.8 to 459.3 and 723.5 pg/mL for S25 and S100, respectively. S25 and S100 concentrations also showed significant differences (*p* = 0.0001) at the end of differentiation (96 h).

### 2.4. Cytokine Response to Sucralose Stimuli on Adipocyte Differentiation

The evaluation of inflammatory cytokines in response to sucralose stimuli at 48 and 96 h of adipocyte differentiation is shown in Figure 4. Results for IL-6 in early-differentiation (48 h) show sucralose decreased the concentrations of S25 by 48% (statistical significance *p* = 0.03) and S100 by 25%, concerning NS (312.5, 162, and 232 pg/mL for NS, S25, and S100, respectively). At 96 h, the same effect was observed with S25, which reduced IL-6 production by 85% (statistical significance *p* = 0.008) and S100 by 18%, to NS (311, 45, and 253 pg/mL for NS, S25, and S100, respectively). Statistical differences were found between S25 and S100 (*p* = 0.01). S25 had a lower concentration of IL-6 at both times of adipocyte differentiation (Figure 4A).

For IL-1β, all the stimuli have similar concentrations of around 200 pg/mL without statistical differences, except for S25 at early differentiation (48 h), which reduces the concentration respect NS by 42% (134.2 and 232.7 pg/mL, respectively, *p* = 0.001) (Figure 4B).

The pattern of IL-8 at 48 h of differentiation showed values of 18,994, 18,363, and 44,158.6 pg/mL for NS, S25, and S100, respectively, without statistical significance. However, at 96 h, NS and S100 remained at 36,000 pg/mL, while the stimulus S25 decreased 7-fold of the concentration respect NS control (5093 pg/mL, statistical difference with NS and S100, *p* =0.02) (Figure 4C).

For TNF-α, it was observed that sucralose decreased the concentration at both differentiation times. TNF-α values at 48 h were 112.7, 49, and 83 pg/mL for NS, S25, and S100, respectively. The S25 stimulus showed a statistical difference with NS control (*p* = 0.009), decreasing 56%. A similar patron was observed at 96 h; the stimulation of S25 significantly reduced the production of this cytokine by 70% to 30.2 pg/mL (*p* = 0.01), while S100 reduced 20% to 83.2 pg/mL (not statistical difference), respect NS control (30.2 pg/mL) (Figure 4D).

For IFN-γ, we do not observe any concentration modification for sucralose stimuli at early or complete differentiation. At 48 h, the concentrations were 913, 866, and 868 pg/mL for NS, S25, and S100, respectively, with no statistical difference. Furthermore, these values were maintained at 96 h, where, despite a 38% decrease with the S100 stimulus, no statistical differences were found between any stimuli (Figure 4E).

Contrary to IFN-γ, the effect of sucralose on the production of IP-10 was evident at 48 h. NS and S100 did not show variation in their concentration with 1600 pg/mL; however, S25 is 38% significantly lower than the NS control and S100 (1028 pg/mL, *p* = 0.01). At 96 h, no statistical differences were found between stimuli. The concentrations were 1503.7, 1324, and 1028 pg/mL for NS, S25, and S100, respectively (Figure 4F).

Finally, regarding IL-10, the effect of S25 and S100 was similar at both differentiation times. At early differentiation (48 h), sucralose stimuli significantly reduced the IL-10 production, around 65% for S25 (12.5 pg/mL, *p* = 0.002) and 44% for S100 (20.2 pg/mL, *p* = 0.01), respect NS control (36.22 pg/mL). In turn, at complete differentiation (96 h), the IL-10 values were 33.1 pg/mL, 15.6, and 15.3 pg/mL for NS control, S25, and S100, respectively; sucralose also significantly decreased IL-10 production by around 52% in the respect NS control, *p* = 0.03 for S25 and *p* =0.04 for S100 (Figure 4G).

## 3. Discussion

The effects of sucralose have been extensively studied in various animal models, primarily murine models. Significant findings from these studies indicate adverse effects such as weight gain, gut microbiota alterations, and the ability of sucralose to cross the placental barrier, leading to concomitant effects [11,14,15]. These results raise serious concerns about the potential health implications of sucralose consumption. Diverse studies also reported cellular damage at sucralose concentrations of up to 1.1 and 11.0 mg/kg/d. For instance, rats fed 3 mg/kg/d showed hepatocytes damage, including nuclear alterations and core destruction (pyknotic nucleus). Besides the hyperplasia of the Kupffer cells, hepatic dysfunction is also a result of sucralose consumption [16]. In addition, elevated concentrations of sucralose ranging from 0 to 3000 mg/kg/d in 26-week-old female rats resulted in alteration of thymus cells and histopathological changes in the cecum [17].

Reports show that sucralose at >10 mM evokes damage at cell turgor on Caco-2, HT-29, and HEK-293 cell lines, making the cell flat and damaging the DNA, showing that colon cells were the most susceptible to the cytotoxic effects of sucralose [10]. Additionally, the embryonic mice hypothalamic cell line, mHypoE-N 43/5, exhibits cytotoxic effects also with 10 mM of sucralose 6 × 10^5^ cells incubated by 48 h has an increase in lactate dehydrogenase (LDH); up to 20 mM, the cell numbers were meager. Also, the amount of LDH was proportional to the necrosis rate [8].

In our work, the sucralose concentrations were used below the reported cytotoxic concentrations, and no cytotoxic effect was observed with any sucralose concentration used. However, it is important to note that these findings should be interpreted with caution, as the effects of sucralose may vary depending on the context and the specific conditions of the study. Our findings on the effect of sucralose on adipocyte differentiation are significant. We evaluated the mRNA expression of gene markers such as CEBPα and PPARγ to white adipocytes, EBP-2, and UCP-1 to brown adipocytes, and acetyl-CoA carboxylase (ACC) as a lipogenesis regulator. CEBPα and PPARγ are synergic transcriptional factors indispensable to white adipogenic differentiation [18,19]. Additionally, CEBPα and PPARγ increased at complete adipocyte differentiation, similar to that observed on adipocyte differentiated to human adipogenic mesenchymal cells due to an increase in reactive oxygen species at the first 72 h with a stimulus of 0.2 mM of sucralose [5]. The same effect was observed in the adipocyte differentiation process on a mouse 3T3 L1 preadipocyte cell line, stimulated with a sucralose concentration of 200 nM for 8 days, where the expression of pro-adipogenic genes as CEBPa markedly increased [11]. These findings suggest that sucralose may significantly impact adipocyte differentiation and gene expression, potentially influencing future research and public health policies.

Our genetic expression results for beige adipocyte markers showed low mRNA levels for UCP-1 and EBF-2 transcriptional factors. This may be attributed to the origin of our cell line, which primarily consists of white adipocytes from subcutaneous adipose tissue; however, beige adipocytes are also present, albeit to a lesser extent [20]. UCP-1 expression increased by almost 140-fold during beige adipocyte differentiation compared to white adipocytes. Garcia et al. (2016) confirmed this increase in gene expression when beige adipocyte differentiation was induced using 1µM of rosiglitazone, a method commonly used to induce beiging in adipocytes in vitro [21]. The low expression level of UCP-1 observed in our study seems to be related to the transdifferentiation process, where white adipocytes can undergo bidirectional conversion into beige adipocytes, a phenomenon known as paucilocular adipocytes [22]. As our results showed, these adipocytes are favorable to the expression of UCP-1, so we believe that sucralose possibility can promote this transdifferentiated phenome in the human adipocyte differentiation process, but it remains unclear [20]. Also, the levels of expression for EBF-2 in our work appear to explain the fact that sucralose can reprogram white adipocytes to beige because white adipocyte EBF-2 is usually inhibited by the zinc finger protein ZBP423, which generally maintains the white adipocyte identity; it has been reported that EBF-2 expression is regulated by UCP-1 expression during beiging of adipocyte [23,24,25].

The activation of PPARγ and CEBPα regulates the expression of lipogenic enzymes, such as ACC, during white adipocyte differentiation [26]. In our study, the ACC mRNA levels increased with the lower concentration of sucralose. A similar result was observed when 0.2 mM and 10 mM sucralose promoted adipogenesis on differentiated adipocytes mesenchymal cells and increased lipid droplets within 72 h of exposure to the nonnutritive sweetener [5]. Previous work in mice showed that 6.4 mg/kg/day of sucralose during pregnancy and lactation increased the diameter adipocyte; it also demonstrated the in vitro effect of sucralose using 3t3-L1 preadipocyte cell line stimulated with 200 nM of sucralose, concluding that sucralose enhances adipogenesis an early stage of differentiation [11]. Our findings agree with the report by Simon et al. (2013) which demonstrated that other stimuli with other nonnutritive sweeteners such as saccharin [2 mM] and aspartame [4.5 mM] also stimulate adipogenesis and alter lipolysis in human and mice cell lines [27].

We found that sucralose seems to alter the production of adipokines, affecting the adipocyte differentiation process. In other work, adiponectin mRNA levels increased 1.5 folds on adipocytes stimulated with sucralose, impairing adipocytes [11]. In obese subjects, the response to supplementation with 0.0024 g/day of sucralose for 12 weeks promotes the synthesis of leptin, adiponectin, and resistin at a concentration of 2.7, 3.4, and 8.5 ng/mL, respectively. On the other hand, the same authors report levels of TNF-α and MCP-1 between 13.18 and 23.71 ng/mL; all these adipokines are related to the development of chronic inflammation and metabolic disorders during obesity [28]. In vivo conditions, several adipokines had pleiotropic effects on tissues and were related to insulin resistance and type 2 diabetes mellitus [20]. With these in vitro results, it seems that the S25 stimulus impacts the expression of essential genes in the white adipocyte differentiation process, which is the majority of visceral and subcutaneous adipose tissue cells. On the contrary, at the protein level, the stimulation with S25 compromises the synthesis of inflammatory cytokines and adiponectin’s characteristic of adipose tissue. Both types of molecules are involved in the inflammatory response on adipose tissue.

However, our work only focused on sucralose impact on the first 96 h of the adipocyte differentiation process; it is necessary to further investigate the effects of sucralose on the whole adipocyte maturation process. We also focused on protein levels in our work, but it was also challenging to determine the physiological concentration at the in vitro level. This opens a wide scope for future research in this area.

## 4. Materials and Methods

The present study was approved by Research Ethics Committee of the Faculty of Medicine, Autonomous University of Sinaloa, with registration number CONBIOÉTICA-25-CEI-003-20181012, and it was conducted under the principles in the Helsinki Declaration.

### 4.1. Cell Culture

ATCC ^®^ Normal human Preadipocytes Cell line PCS 210-010 were grown on fibroblast growth medium supplemented with 2% FSB (PCS-201-030) and fibroblast grown kit (PCS-201-041) which contains: L-glutamine: 7.5 mM, rh FGF basic: 5 ng/mL, rh Insulin: 5 µg/mL, Hydrocortisone: 1 µg/mL, Ascorbic acid: 50 µg/µL at 37 °C and 5% of CO_2_, until a confluence of 80%. The adipocyte differentiation was performed using the ATCC PCS-500-050 kit, which contains adipocyte basal medium and AD Supplement to initiate the differentiation, and the ADM supplement to maintain the adipocyte differentiation.

### 4.2. Cell Differentiation

Two concentrations of sucralose S25 and S100 (25 ng/µL and 100 ng/µL, respectively) were evaluated during the adipocyte early differentiation process at 48 h and the complete differentiation process at 96 h. In previous experiments, viability tests were performed at concentrations of 6.25, 12.5, 25, 50, and 100 µg/µL, obtaining cytotoxic effects in all cases. For the present work, we decided to evaluate the concentrations of 25 and 100 ng/µL, with no cytotoxic effects. The diluent but without sucralose (NS) was used as a control. The sucralose was 99% pure, NuSci^®^ CAS: 56038-13-2, diluent on sterile milli Q water. 40,000 cells were seeded in 12-well plates with basal fibroblast growth medium supplemented with 5% FBS for 48 h under culture conditions at 37 °C with 5% CO_2_. The basal fibroblast growth medium was removed, and the cells were washed at least twice with tempered sterile 1X PBS. The culture was resuspended on 2 mL of initiation medium AD with sucralose. It was incubated under the same culture conditions, replacing the media every 48 h and finalizing the differentiation process at 96 h. All the experiments were performed in triplicate on both sucralose concentrations (S25 y S100), and the supernatant was collected at 0, 48, and 96 h of differentiation.

### 4.3. Morphology of Differentiated Adipocytes

The morphology of the differentiated cells was documented using an inverted microscope (Carl ZeissTM Axio Vert.A1, ZEISS, Oberkochen, Deutschland) coupled to a microscope camera (Axiocam ERc 5s, ZEISS, Oberkochen, Deutschland) using the AxioVision Sofware 4.8 version. The images were documented with the 20X objective.

### 4.4. Isolation of RNA and cDNA Synthesis

According to the manufacturer instructions, total RNA isolation was carried out using TRIzol reagent (Zymo Research, Irvine, CA, USA, #R2050-1-200), concentration and quality were assessed by Nanodrop spectrophotometer, and integrity by 1% agarose gel. Then, single-stranded cDNA was synthesized using 1000 ng of total RNA in a final volume of 20 µL, using RevertAid H Minus First Strand cDNA Synthesis Kit (Thermo Scientific, Waltham, MA, USA, #K1632) and oligo dT. The reaction mixture was incubated at 25 °C for 10 min, 42 °C for 60 min, and 70 °C for 10 min.

### 4.5. Gene Expression Analysis

Specific primers were designed with Primer3 Software version 4.1.0 (https://primer3.ut.ee/ access date 13 December 2024 to PPAR**γ**2 (Fw 5′-ACAGATCCAGTGGTTGCAGA-3′ Rw 5′-ATGAGGGAGTTGGAAGGCTC-3′), EBF-2 (Fw 5′-GGGTCCGGAATGTCGGAG-3′ Rw 5′-CCACAAAGTCCACGAAGGC-3′), ACOA (Fw 5′-GCACATAAGGTCCAGCATG-3′ Rw 5′-GCGAGTAACAAATTCTGCTGG-3′), and GAPDH (Fw 5′-CTGCACCACCAACTGCTTAGCG-3′ Rw 5′-TCA TGT TCT GGA GAG CCC CG-3′) genes, and C/EBP-α (Fw 5′-CAAATATTTTGCTTTATCAGCCGATA-3′ Rw 5′-CGCACATTCACATTGCACAA-3′) and UCP-1 (Fw 5′-GTGTGCCCAACTGTGCAATG-3′ Rw 5′-CCAGGATCCAAGTCGCAAGA-3′) primers were taken from previous reports [29,30]. Real-time PCR was performed on thermocycler MyGo Pro from IT IS Life Science LTD-2019 (Dublin, Ireland) using 2X Fast Start SYBRGreen Master (ROX) from ROCHE^®^ (Darmstadt, Germany), 10 pmol for each primer set and 100 ng de cDNA in 10 µL of final reaction volume. The qPCR conditions were as follows: 10 min at 95 °C, 40 cycles of 15 s at 95 °C, and 1 min at 60 °C. Fold changes were calculated using the 2^−ΔΔCt^ method with GAPDH data expression as a normalization control.

### 4.6. Quantitation of Human Panel Adipokine and Cytokine by Flow Cytometry

Cytokines were measured from the collected supernatants at 0, 48, and 96 h after differentiation using the Human Adipokine Panel (13 plex), LEGEND plex (TM) Multi-analyte Flow assay kit, Cat. No. 740196 (BioLegend, San Diego, CA, USA), by flow cytometry (BD FACS Canto II, (BD Bioscience, Franklin Lakes, NJ, USA), following the manufacturer’s instructions. Briefly, 25 µL samples and standards were loaded on glass tubes, followed by the addition of premixed beads. The plate was then covered and incubated for 2 h at room temperature (RT) on a plate shaker at 800 rpm. The tubes were centrifugated by 5 min at 1050 rpm. After two washing steps with 200 µL of wash buffer, detection antibodies were added and incubated for 1 h at RT on a plate shaker at 800 rpm. 25 µL of SA-PE was added directly after incubating and incubated on a shaker at 800 rpm for 30 min at RT. The last washing step was performed, and a flow cytometer analyzed the samples. Data analysis was performed with Data Analysis Software Suite for LEGENDPLEX TM cloud version available on https://www.biolegend.com/en-us/immunoassays/legendplex/support/software accessed on 13 December 2024. available on accessed on 13 December 2024

### 4.7. Statistical Analysis

The statistical analysis was performed using one-way ANOVA and Tukey’s multiple comparison test as a post hoc test to determine the statistical differences in the genetic expression of differentiation markers. Differences between cytokine productions on differentiation times and sucralose concentration were evaluated for significance using the two-way ANOVA and Tukey’s multiple comparison test as a post hoc test. In both analyses, a *p*-value of 0.05 was considered significant. The data are presented as mean ± Standard Deviation (SD) of the mean of independent measurements. All the statistical analyses were performed using GraphPad PRISM 8 Software.

## 5. Conclusions

In conclusion, the nonnutritive sweetener sucralose has been found to impact adipocyte differentiation in the early stages of the process. This effect is evidenced by an increase in the genetic expression of CEBP-α and ACC synthase, both related to lipogenesis. Furthermore, sucralose affects the synthesis of inflammatory adipokines and cytokines, showing decreasing levels of adiponectin and MCP-1, which may disrupt normal adipocyte function. However, its impact on mature adipocytes and adipocyte synergy is yet to be determined. This research is crucial as it can potentially shed light on the effects of nonnutritive sweeteners on metabolic response and their role in developing obesity and metabolic syndrome, thereby influencing public health. Further investigations are necessary to understand these effects fully, underscoring the importance of continued work in this area.

## Figures and Tables

**Figure 1 ijms-25-13635-f001:**
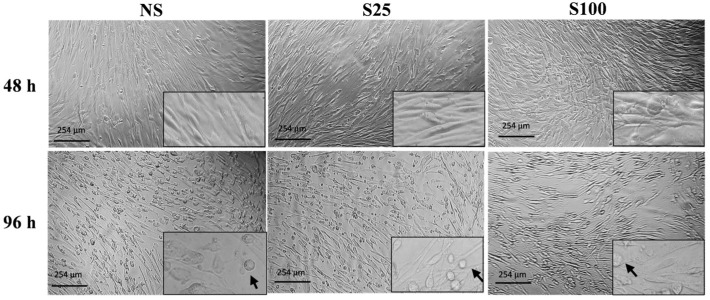
Morphology of adipocytes stimulated with sucralose in the differentiation process. Differentiation process at 48 h and 96 h on preadipocytes stimulated with sucralose. NS (control without sucralose), S25 (25 ng/µL sucralose stimulus), S100 (100 ng/µL sucralose stimulus). The black arrows in the box represent normal adipocyte morphology.

**Figure 2 ijms-25-13635-f002:**
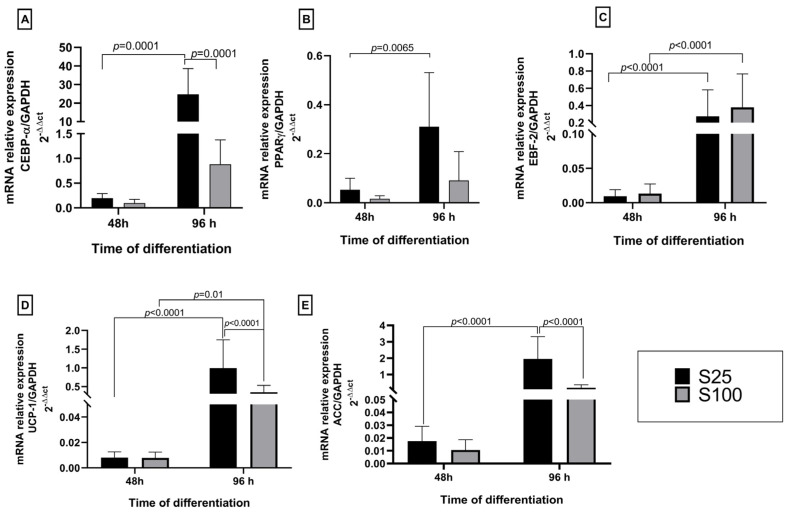
Genetic expression of differentiation markers for white and brown adipocytes and regulator of lipogenesis. Expression of differentiation markers for white adipocytes (**A**,**B**). Expression of differentiation markers for brown adipocytes (**C**,**D**). Genetic expression of lipogenesis regulator in (**E**). The stimuli evaluated were sucralose 25 ng/µL (S25) and 100 ng/µL (S100). The statistical significance was in the performance of one-way ANOVA and Tukey’s multiple comparisons test as a post hoc test. The statistical analysis shows the mean and standard deviation using GraphPad PRISM 8 software.

**Figure 3 ijms-25-13635-f003:**
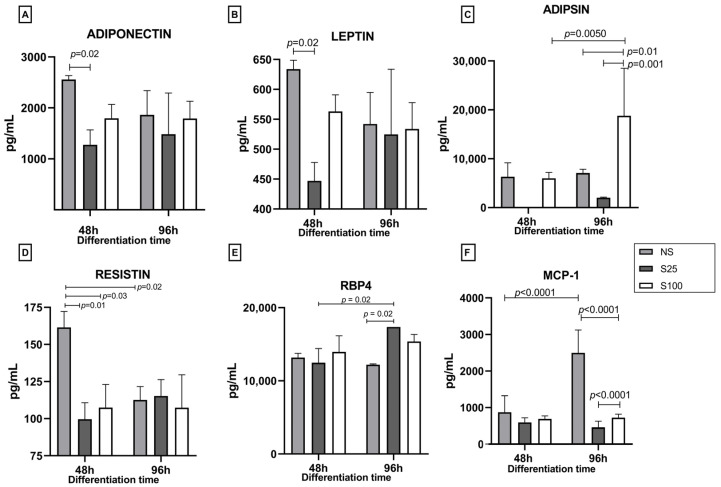
Quantitation of human adipokine in response to sucralose stimuli by flow cytometry. Adipokine concentration on the supernatant of adipocyte cell culture during the adipocyte differentiation process. Early differentiation is determined at 48 h and complete differentiation at 96 h. The stimuli evaluated were control of differentiation (NS), sucralose 25 ng/µL (S25), and 100 ng/µL (S100). Adiponectin (**A**), Leptin (**B**), Adipsin (**C**), Resistin (**D**), RBP4 (**E**), and MCP-1 (**F**). The statistical significance was performance using two-way ANOVA and Tukey’s multiple comparisons test as a post hoc test. The statistical analysis shows the mean and standard deviation using GraphPad PRISM 8 software.

**Figure 4 ijms-25-13635-f004:**
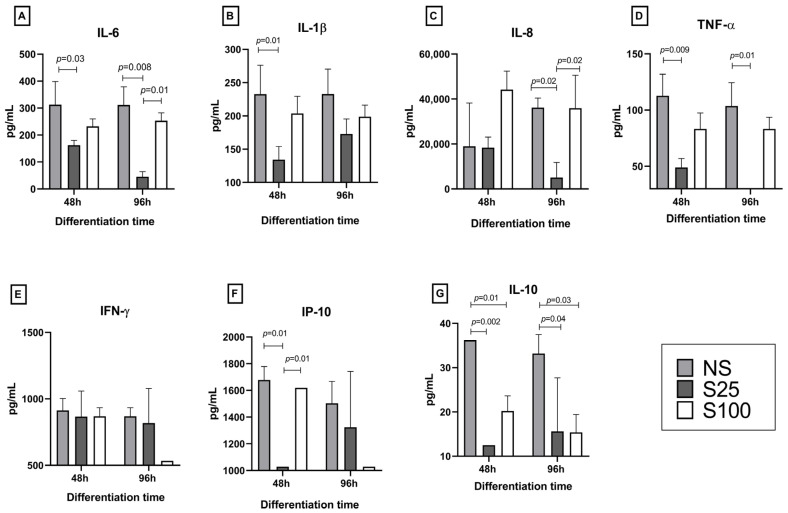
Effect of sucralose on cytokine response during adipocyte differentiation. Cytokine concentration was detected on the supernatant of adipocyte cell culture stimulated with sucralose during the adipocyte differentiation process. Early differentiation is determined at 48 h and complete differentiation at 96 h. The stimuli evaluated were control of differentiation (NS), sucralose 25 ng/µL (S25), and 100 ng/µL (S100). IL-6 (**A**), IL-1β (**B**), IL-8 (**C**), TNF-α (**D**), IFN-γ (**E**), IP-10 (**F**), and IL-10 (**G**). The statistical significance was performance using two-way ANOVA and Tukey’s multiple comparisons test as a post hoc test. The statistical analysis shows the mean and standard deviation using GraphPad PRISM 8 software.

## Data Availability

Data are contained within the article.

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
