# Peer review of "Sucralose-Enhanced Adipogenesis on Preadipocyte Human Cell Line During Differentiation Process"

_ijms, 2024, doi:10.3390/ijms252413635_

Round 1
Reviewer 1 Report
Comments and Suggestions for Authors
In the following manuscript Authors analyzed the effect of sucralose, a commonly non-nutritive sweetener, on the differentiation process of adipocytes to determine whether sucralose can promote adipogenesis. Sucralose at concentrations of 25 (S25) and 100 ng/uL (S100) was tested against the control without sweetener (NS) during the differentiation process of adipocytes at 48h and 96h.Genetic expression of markers of adipogenesis was evaluated. In addition, human cytokines related to the inflammatory response were measured in the medium. From the results obtained authors affirm that sucralose induces increased expression of the white adipocyte differentiation marker CEBP-α and the lipogenesis regulator ACC at 96 h before complete differentiation. In conclusion authors affirm that sucralose stimulates the expression of adipogenesis-related genes and negatively influences the secretion of inflammatory cytokines and adipokines during preadipocyte differentiation.
Some points need to be revised:
Protein expression analysis by Western Blot of the analyzed genes by qPCR would be appropriate.
Phenotypic analysis by Oil red Assay analysis of lipid accumulation by ImageJ would be helpful.
Also, how did Authors decide on the concentrations of sucralose to be used? Did you do trials with intermediate concentrations between 25 and 100? Was a cytotoxicity evaluation of these two concentrations done before testing?
In addition, I would suggest to analyzed gene expression by qPCR of cytokines detected in the medium.
Reviewer 2 Report
Comments and Suggestions for Authors
Manuscript titled “Sucralose enhanced adipogenesis on preadipocyte human cell line during differentiation process” reports an in vitro analysis of the effects of two concentrations of sucralose on adipocyte differentiation and inflammation markers. The work is interesting, although there are some comments and suggestions for the authors:
1. Section 4.2 states that 25 and 100 ng/uL were used. Can you please justify these specific concentrations? Were they selected based on the literature or on previous experiments?
2. Line 353 states that “primers were taken from previous reports”. Please add the appropriate citations for the primers used.
3. Line 95 mentions a “statistical difference”, but references a p value of “.065”. Did you intend to say “no statistical difference” or should the p value be a different value instead? Figure 2B appears to confirm the 0.065 value, but please check and confirm this sentence.
4. The graphs shown in Figure 2 are too small and have too much blank space between them. Please consider increasing their size and/or eliminating the blank space, in order to make them easier to read. Similar comment for Figure 3.
5. Lines 234-235 state that “However, this is not like other authors' findings because the sucralose at concentrations above 10 mM markedly affects…”. In this reviewer’s opinion, your findings are not different from those of the literature (at least according to this statement), since you used concentrations well below the 10 mM used by other authors.
6. In line 241, please specify the acceptable daily intake value mentioned here. Moreover, since your model did not “intake” the compound, how does a daily intake value intended for an adult to consume compare to the 25 and 100 ng/uL added in vitro to a cell line? In other words, how does the acceptable daily intake relate to the doses administered in the present work?
7. Please specify to which cell lines the results mentioned in lines 231-232 refer to.
8. In line 293, what does “am” mean?
9. The conclusion mentions that sucralose “impair adipocyte differentiation”, however, this idea was better stated in line 256 “sucralose may significantly impact adipocyte differentiation” (replacing “impair” with “impact”); please consider using this wording instead. Moreover, this phrase may be better substantiated by mentioning which changes were found that support it, for example, “sucralose may significantly impact adipocyte differentiation, according to an increase in X gene expression”. Similar comment for the phrase “negatively affects the synthesis…”, which could be better supported by mentioning which changes led to this conclusion.
10. There are some minor typos, including in line 71 (should it be “higher” instead of “mayor”?), line 312 (should it be “FBS” instead of “SFB”?), line 238 (should it be “10 to the five” instead of “105”?) and in line 345 (do you mean “Primer3” instead of “Primer out 3”?). Please carefully check your document for any other minor mistakes that could be present.
Author Response
"Please see the attachment"

Round 2
Reviewer 1 Report
Comments and Suggestions for Authors
I thank authors for comments and clarifications
Author Response
there are no comments to this reviewer at round 2
Reviewer 2 Report
Comments and Suggestions for Authors
Manuscript titled “Sucralose enhanced adipogenesis on preadipocyte human cell line during differentiation process” reports an in vitro analysis of the effects of two concentrations of sucralose on adipocyte differentiation and inflammation markers. The present version of the manuscript was modified according to comments and suggestions made during an initial review, those made by the present reviewer include:
1. Justifying the authors’ choice of using 25 and 100 ng/uL doses. The authors comment on their response letter that their choice was based on previous experiments, however, this is not explained to the reader in section 4.2. Please briefly mention your rationale to the reader.
2. Adding the corresponding references from where their primers were taken. The authors added references 30 and 31.
3. Confirming a “statistical difference”, and its corresponding p value of “.065”. The authors corrected to “0.0065”.
4. Resizing figures 2 and 3 for improved readability. Both figures were adequately modified.
5. Restructuring a sentence to agree with the literature, regarding the treatments’ cytotoxicity. The sentence was modified accordingly.
6. Clarifying how a dose administered to a cell line compared to the daily intake established for a human. The authors restructured the corresponding sentence to omit this comparison.
7. Specifying to which cell lines the results mentioned in lines 231-232 refer to. The authors restructured the sentence to specify the appropriate cell lines.
8. Confirming an abbreviation in line 293. The word was deleted.
9. Improving some phrasing in the conclusion and specifying which results led to that particular statement. The conclusion was rephrased accordingly.
10. Finally, correcting some minor typos throughout the manuscript. The typos were corrected.
According to the aforementioned changes made by the authors, it appears that they adequately considered and addressed the present reviewer’s comments and suggestions. The only minor issue that remains is to clarify to the reader their choice of doses administered (comment 1), however, this change does not merit another review round.
